# The Effect of Drying Methods and Extraction Techniques on Oleuropein Content in Olive Leaves

**DOI:** 10.3390/plants11070865

**Published:** 2022-03-24

**Authors:** Darija Cör Andrejč, Bojan Butinar, Željko Knez, Kaja Tomažič, Maša Knez Marevci

**Affiliations:** 1Faculty of Chemistry and Chemical Engineering, University of Maribor, SI-2000 Maribor, Slovenia; darija.cor@um.si (D.C.A.); zeljko.knez@um.si (Ž.K.); kaja.tomazic@hotmail.com (K.T.); 2Institute for Oliveculture, Science and Research Centre Koper, SI-6000 Koper, Slovenia; bojan.butinar@zrs-kp.si; 3Faculty of Medicine, University of Maribor, SI-2000 Maribor, Slovenia

**Keywords:** olive leaves, drying, different extraction techniques, oleuropein

## Abstract

Increased demand for olive oil has caused higher quantities of byproducts in olive processing, such as olive leaves, olive skins, and vegetation water. It is well known that olive leaves contain several phenolic compounds, including secoiridoids. Oleuropein is the major secoiridoid in olive leaves. Oleuropein has been found to exhibit antioxidative, antimicrobial, antiviral, and antiatherogenic activities. We studied the effect of extraction techniques and drying methods on oleuropein content in olive leaves of *Istrska belica* and *Lecino* cultivar. Three different procedures of drying were used: at room temperature, at 105 °C, and freeze drying. Ethanol-modified supercritical extraction with carbon dioxide, conventional methanol extraction, and ultrasonic extraction with deep eutectic solvent were performed. Antioxidant activity was determined, as well as methanolic and supercritical extracts. The presence of olive polyphenols was confirmed by the HPLC method.

## 1. Introduction

During olive oil production, the crude olive cake, twigs, and vegetation water are the main byproducts of olives. In the olive industry, olive leaves represent around of 10% of the total olive weight. Some amount of leaves can be found during pruning of olive trees [1,2,3]. Applications of olive leaves in the industry are limited. Large quantities of olive leaves are deposited in nature or removed by incineration, which is potentially harmful to the environment; therefore, it makes sense to explore new possibilities for the use of processing residues from olive growing.

Olive leaves are most used in animal feed. A small amount of leaves (around 2–3%) can be mixed with olives before oil processing to produce more marked flavor and produce a product a greater resistance to oxidation [4]. Moreover, they are used in the form of liquid extracts and powders as food additives and functional food materials without complete chemical characteristics [4,5,6]. Because of natural sources of bioactive compounds (especially phenols), olive leaf extracts can be used in the food additive industry, as well as the cosmetics and pharmaceutical industries.

Olive leaves are known to have many useful pharmacological effects. Some of their health benefits are related to phenolic composition, especially to oleuropein (OLE) and flavonoid content. Olive leaves contain many phenolic compounds, such as secoiridoids, with a high molecular weight of up to 600, containing one or two hydroxy aromatics, which are connected to other non-aromatic components [7]. The major secoiridoid in olive leaves is oleuropein, a heterosidic ester of hydroxytyrosol with β-glucosylated elenolic acid [8,9]. Pereira et al. identified seven phenolic compounds in in olive leaf extract: caffeic acid, verbascoside, oleuropein, luteolin 7-O-glucoside, rutin, apigenin 7-O-glucoside, and luteolin 4′-O-glucoside. It was found that the aqueous extract showed a profile in which oleuropein was presented in the highest amount, ca. 73% of total identified compounds [10]. The health benefits associated with the consumption of oleuropein include the prevention of cardiac diseases, improvement in lipid metabolism, and decrease in obesity-related diseases, among others. Oleuropein and its metabolite, hydroxytyrosol, have powerful antioxidant activity, which might be responsible for some of olive oil’s antioxidant, anti-inflammatory, and disease-fighting activities. Oleuropein is well known for its blood-pressure lowering effect. It has been found to protect the hypothalamus from oxidative stress by improving mitochondrial function through activation of the Nrf2-mediated signaling pathway [11]. Besides hypertension, oleuropein has been shown to have cardioprotective, anti-inflammatory, antioxidant, anti-cancer, antiangiogenic, and neuroprotective functions [4,12,13,14,15]. Oleuropein was reported to reduce oxidative damage in aged rat brains that had been affected by Parkinson’s disease [16]. Moreover, it prevents the toxic aggregation of both amyloid beta and tau, proteins that are involved in Alzheimer’s disease [17,18]. Oleuropein is also a potent inhibitor of human epidermal growth factor receptor 2, which is mostly overexpressed in breast cancer cells. Menendez et al. reported that oleuropein synergizes with trastuzumab-sensitive breast cancer cell lines [19].

To obtain olive leaf extract with a high content of oleuropein, special attention should be given to the choice of extraction method. In majority of cases, the industrial processes for obtaining olive leaf extract involves the use of organic solvents, which present a serious threat to human health and the environment. Application of supercritical (SC) fluids for the extraction of high-quality extracts without organic solvent residues is thus a very promising alternative. Carbon dioxide is the most frequently used SC fluid and is environmentally friendly and cheap.

Eutectics are modern solvents made from cheap, non-toxic, and recyclable components [20]. They are characterized by their ease of synthesis, biodegradability, and non-flammability [21]. They are formed by mixing two (or more) components in a solid or liquid aggregate state, which, in each molar ratio and at a given temperature, form a liquid with has a significantly lower melting point than that of the individual components [22]. The key to the formation of deep eutectic solvents (DESs) is the hydrogen-bonding interactions between the hydrogen bond donor (hydrogen bond donor, HBD) and the hydrogen bond acceptor (hydrogen bond acceptor, HBA) [23]. Strong hydrogen bonds in eutectic solvents result in good miscibility with other solvents, especially water, and high extraction power for extracting active substances from biomass [20]. In the present work, the simultaneous influence of different drying methods and extraction procedures on the active substance content in olive leaves was examined. We studied the differences in the content of active phenolic compounds in the leaves of two olive cultivars: *Istrska belica*, which is the most represented cultivar in Slovenian olive groves; and *Leccino*, which was brought to Croatian Istria after 1960, from where it later expanded into Slovenian Istria. The *Leccino* cultivar tolerates low temperatures well and recovers rapidly after frosts. Compared to *Istrska belica*, it is more resistant to diseases and pest attack [24,25]. Different extraction methods were performed using different solvents in order to obtain high-phenolic-content extracts. We examined the use of an ecofriendly DES as a highly effective solvent for the extraction of phenols from olive tree leaves. In the final stage, the extracts were evaluated according to the content of phenolic compounds determined by the HPLC method.

## 2. Results

### 2.1. Results of Drying Olive Leaves (Air Drying at Room T, at T = 105 °C, and Freeze Drying)

Drying is the oldest method of preserving leaves. Immediate drying of olive leaves is the most important stage of post-harvest processing, as it prevents a significant reduction in quality and degradation during storage. In addition, leaves must be dried before extraction to reduce moisture content. Fresh leaves contain free and bound water, which must be removed to prolong shelf life. The presence of moisture causes enzymatic and microbial activity that leads to a reduction in nutritional value and degradation. Drying is also economical, as the leaves lose up to 85% of their weight, which reduces storage and transport costs [26].

We performed three different drying methods for two cultivars of olive leaves (*Istrska belica* and *Leccino*): air drying at room T in a dark place, air drying at 105 °C for 90 min, and freeze drying-lyophilization. The results of water content in dried olive leaves are presented in Table 1, and the average value of three parallels is given. According to the European Pharmacopoeia, monograph 1878 on olive leaves, the moisture content is less than 10.0%, related to the conditions described in the European Pharmacopoeia 10.0. [27]. On the other hand, the moisture content in our samples was between 3.4% and 6.37% and was highest after freeze drying.

### 2.2. Results of Ethanol-Modified Supercritical Extraction of Olive Leaves

Six supercritical extractions were performed using carbon dioxide (CO_2_) and ethanol as a cosolvent. Extraction was carried out at a pressure of 235 bar and a temperature of 40°C. An HPLC pump was used to ensure a constant flow of ethanol (1 mL/min) during extraction. The solvent extract was collected in tubes. Afterwards, the solvent was evaporated, and yield was calculated. In the case of samples of *Istrska belica*, extraction was carried out by collecting fractions every 30 min, whereas in the case of *Leccino*, the extraction took place in one cycle of 120 min. Results are presented in Table 2.

Besides the low solvent polarity, low extraction temperature may have also contributed to the low extraction yield. The highest yield, approx. 12.6%, has was attained by *Istrska belica*. In the case of *Istrska belica*, the highest extraction yield was achieved in the case of air drying at room temperature, followed by lyophilization and drying at a temperature of 105 °C. Generally, the lowest extraction yield was found in the case of drying leaves at 105 °C for both cultivars.

#### 2.2.1. DPPH Activity of SCE Extracts

The results of DPPH * radical scavenging activities of supercritical extracts of *Istrska belica* are presented in Figure 1. Samples differed according to the method of drying (air—room temperature, dryer—air (T = 105 °C), lyophilization) and fractions obtained at 30 min, 60 min, 90 min, and 120 min. Inhibition of *Istrska belica* extracts varies between 49% and 98%, depending on the time of extraction and type of drying. The highest DPPH activity was determined in extracts of leaves dried in a freeze dryer, followed by extracts obtained from leaves dried in a dryer (T = 105 °C), and the lowest antioxidant activity in air-dried leaves. It can be seen from the Figure 2 that *Leccino* extracts had lower DPPH activity than *Istrska belica.*

#### 2.2.2. HPLC–AD Analysis of SC Extracts

The content of active substances in olive leaves was determined by the HPLC–DAD method. Oleuropein was the main compound we focused on. The results are given as the mass of active compound in mg per g dry weight (d.w.) (mg_c_/g d.w.). Components such as hydroxytyrosol (TyrOH), hydroxy oleuropein (OH-Ole), verbascoside (Ver), luteolin-7-glucoside (Lu-7-O-Glu), oleuropein (OLE), ligstroside (Lig), and oleuroside (Ols) were successfully characterized in the extracts. The results of the HPLC analysis of several active compounds for *Istrska belica* and *Leccino* are shown in Table 3 and Table 4, respectively.

From Table 3, it can be observed that the highest oleuropein content was achieved by freeze drying, followed by air drying at room T and air drying at T = 105 °C. In general, the highest oleuropein content for *Istrska belica* was achieved in the case of 90 min of supercritical extraction, regardless of the type of leaf drying. The reason for this is most likely the nature of the SCE process, wherein fewer polar compounds elute first. The highest content of oleuropein was achieved in leaves dried by freeze dying, namely 13.60 mg/g of dry leaves. In smaller quantities, we found other components that stood out, such as luteolin-7-glucoside (Lu-7-O-Glu), oleuroside (Ols) (up to 1 mg/g), apigenin-7-glucoside (Api-7-O-Glu) (around 0.5 mg/g), and others. In the case of freeze drying, an increase in the concentration of some components, such as Ols, Lu-7-O-Glu, ligstroside (Lig), Api-7-O-Glu, and hydroxytyrosol glucoside (HO-Tyr-O-Glu), was detected in comparison with air drying at room T or air drying at T = 105 °C. For *Leccino*, the highest oleuropein content was achieved in leaves dried by air at room T, followed freeze drying and air drying at T = 105 °C. Samples dried in a freeze dryer differed from other supercritical extracts by their high concentration of luteolin (Lu) of around 0.439 mg/g and apigenin (Api) (1.330 mg/g). However, the values of oleuropein in the *Leccino* variety are significantly lower than those in *Istrska*
*belica*. A comparison of oleuropein content is shown in the Figure 3, where only extract fractions after 90 min are compared.

### 2.3. Results of Methanol Extraction of Olive Leaves

Methanol extraction was performed according to the method of the European Pharmacopeia 10.0 [27]. We studied the influence of the type of drying on the antioxidant activity of the extracts. The phenolic components of the extracts were also determined by HPLC analysis.

#### 2.3.1. DPPH Activity of Methanol Extracts

DPPH activity is presented in Figure 4. *Leccino* extracts have much lower antioxidant activity compared to *Istrska belica*. For *Istrska belica*, the highest values were obtained in the case of freeze drying (around 94%), followed by drying at T = 105 °C (86%) and drying in air, whereby antioxidant activity was around 82%. On the other hand, the highest DPPH activity of 72% for *Leccino* was obtained in the case of air drying at 105 °C.

#### 2.3.2. HPLC–DAD Analysis of Methanol Extracts

Results of HPLC analysis of methanol extracts are presented in Figure 5 and in Appendix A. *Istrska belica* methanol extracts achieved higher oleuropein content than *Leccino* extracts. However, the hydroxytyrosol glucoside (HO-Tyr-O-Glu 1 and 2) and oleuroside (Ols) contents were higher in *Leccino* extracts. In Figure 5, it can be seen that the oleuropein content was highest when leaves were dried with air at room T, followed by air drying at 105 °C for both *Istrska belica* and *Leccino*. Extracts of lyophilized leaves had a significantly lower content of oleuropein compared to other methanol extracts: only about 5 mg/g d.w. in the case of *Istrska belica* and 3 mg/g d.w. for *Leccino*. Based on these results, we can conclude that *Istrska belica* is richer in oleuropein than *Leccino.* Two isomers of methoxy-oleuropein, peaks at the retention time of approximately 32.5 min, marked as methoxy-oleuropein a and b (MeO-OLE a and 2-MeO-OLE b) with a value in *m/z* at 569 were also detected in the methanol extracts of lyophilized leaves. This secoiridoid glucoside was previously reported in other cultivars [28] and for the first time in olive leaves by Taamalli et al. [29]. Talhaoui et al. also detected methoxy-oleuropein and its isomer in ‘Sikitita’ olive leaves [30]. The presence of secologanoside (Sec) was detected only in extracts of freeze-dried olive leaves. Extraction with methanol produced higher oleuropein content in *Istrska belica* extracts, whereas the *Leccino* extracts had higher contents of other phenols.

### 2.4. Results of Ultrasound Extraction with Deep Eutectic Solvent

Deep eutectic solvent (DES) was prepared with glycerol, glycine, and water (Gli:Gly:H_2_O = 7:1:9). Extracts were analyzed with the HPLC method, and the influence of the type of drying on the concentration of phenolic components was studied for *Istrska belica* and *Leccino* cultivars. Results are presented in Figure 6 and in Appendix A.

The high content of phenolic compounds in eutectic extracts of *Istrska belica* and *Leccino* demonstrate that eutectics are also an effective solvent for the extraction of active compounds from olive leaves (Figure 6). The highest amounts of oleuropein in *Istrska belica* extracts were obtained in the case of air drying at room T (up to 45 mg/g d.w.), followed by the samples subjected to air drying at 105 °C (43.1 mg/g d.w.). In the case of *Leccino* leaf extraction, the concentration of oleuropein (OLE) was lower in the case of drying at room T. Figure 6 indicates that the lowest oleuropein concentration was recovered in samples after freeze drying for both cultivars, whereas concentrations for *Istrska belica* and *Lecino* reached only 0.93 mg/g d.w. and 0.27 mg/g d.w., respectively. Caffeoyl-6-secologanoside (Caff-6-sec) was detected in small amounts only in DES extracts of *Istrska belica* and *Leccino* dried at room temperature and at 105 °C. Oleuropein glucoside with *m/z* 701 was also detected in eutectic extracts [22]. Luteolin-7-glucoside represents the highest concentration (between 1.3 and 2 mg/g d.w.) of polyphenols among the extracts from freeze-dried leaves for both *Istrska belica* and *Leccino*. Isomers with *m/z* 612 and *m/z* 723 were also detected in extracts of freeze-dried leaves in very small amounts. When lyophilization was used, the concentrations of active components in extracts were very low compared to extracts obtained by other drying techniques. Chromatograms of *Istrska belica* deep eutectic extracts from DES1–DES3 are presented on Appendix A.

## 3. Discussion

Several studies have been performed about the extraction of oleuropein by different extraction methods. Not so many works have compared drying techniques and extraction methods. Drying of olive leaves before extraction is required to achieve high oleuropein content. Numerous studies have shown that fresh leaves have a lower oleuropein content than dry leaves due to the action of the enzyme β-glucosidase [26]. Afaneh et al. found that drying fresh olive leaves at room temperature is the best method, as it preserves oleuropein from degradation compared to drying at higher temperatures [31]. Shuichi et al. studied the effect of drying olive leaves. They found out that higher temperatures are not suitable, as oleuropein can be degraded the activating enzymes at high temperatures [32]. Şahin et al. studied the effect of drying method on oleuropein content, using freeze drying, vacuum drying, oven drying, and ambient air drying. A significant decrease in oleuropein content was reported using oven drying [33]. Today, freeze drying is considered one of the most suitable and most appropriate methods for preserving a wide range of plant products. Accordingly, we performed three different methods of drying: air drying at room T, hot air drying at T = 105 °C, and freeze drying.

Firstly, the antioxidant activity of extracts was obtained. From the results of DPPH activity of supercritical extracts (Figure 1) it can be seen that the percentage of inhibition increases from fraction 1 to 4, which is attributed to the extraction of polar phenolic components in higher fractions. This refers to the largest proportion of active compounds in olive leaves being polar, so they can be extracted after a long time during supercritical extraction. It can be observed that the inhibition values for *Istrska belica* are much higher than those for *Leccino*. Leaves of *Istrska belica* have a higher antioxidant content, regardless of the drying method used. Antioxidant activity of *Leccino* and *Istrska belica* supercritical extracts are lowest in the case where the leaves were dried by lyophilization. The highest inhibitions of 98%, 86%, 80% were determined in supercritical extracts of freeze-dried *Istrska belica* leaves, leaves dried at room T, and leaves dried hot air drying at 105 °C, respectively. Difonzo et al. demonstrated that pretreatment of leaves affects on both polyphenol content and antioxidant activity [34].

Methanol extraction exhibited a similar trend in antioxidant activity for both cultivars. Higher values of DPPH radical inhibition were obtained in the case of extracts from *Istrska belica* (up to 94%). Kiritsaki et al. extracted olive leaves by using solvents of increasing polarity (petroleum ether, dichloromethane, methanol, and methanol/water: 60/40). The highest DPPH activity of around 95% was determined with methanol:water extraction, followed by methanol extraction (around 92%). These values are comparable with our results for *Istrska belica* methanol extraction [35]. *Leccino* methanol extracts have DPPH inhibition between 48% and 60.9%. In our research, extracts of *Istrska belica* had much higher antioxidant activity than *Leccino*, regardless of the extraction method and solvent (ethanol-scCO_2_, methanol) used. Because the eutectic extracts were obtained as solutions, antioxidant activity was not determined. The identification and quantification of phenolic compounds in *Istrska belica* and *Leccino* leaves were based on their spectra, on their retention time in comparison with phenolic standards analyzed under the same conditions, and on the method of standard addition to the samples. Eluates were detected at 280 nm, as in the research of Buaziz et al.; besides, oleuropein, six flavonoids (luteolin 7-O-glucoside, luteolin 7-O-rutinoside, apigenin 7-O-glucoside, rutin, luteolin, and apigenin) were identified [36]. The main focus of our research was the identification of oleuropein; therefore, all identified compounds were quantified on the basis of oleuropein response at 280 nm. In addition to oleuropein, we successfully identified the polyphenol-like hydroxytyrosol glucoside, methoxy-oleuropein, apigenin, apigenin-7-glucoside, glucoside-oleuropein, hydroxy oleuropein, ligstroside, luteolin, luteolin-7-glucoside, oleuroside, secologanoside, hydroxytyrosol, and verbascoside.

Supercritical extracts of freeze=dried *Istrska belica* leaves have oleuropein content of 13.6 mg/g d.w., followed by air dry leaves extracts at room T, with an oleuropein concentration of 6.8 mg/g d.w. The lowest concentration of oleuropein (around 2.3 mg/g d.w.) was determined in hot air-dried leaf extract. The oleuropein concentration in the *Leccino* supercritical extracts were significantly lower than in *Istrska belica* extracts, regardless of the type of drying. In air-dried *Leccino* extract, the highest amount of oleuropein was determined to be around 3.5 mg/g d.w. Freeze-dried *Leccino* leaf extracts differed from other supercritical extracts in their high concentrations of luteolin (Lu), with 0.439 mg/g d.w., and apigenin (Api), with 1.33 mg/g d.w. The oleuropein content of *Istrska belica* in methanol extracts was 77.7 mg/g d.w. and 70.1 mg/g d.w. for air-dried leaves at room T, and hot air drying at 105 °C, respectively. Oleuropein contents of *Leccino* methanol extract were lower (66.1 mg/g d.w.) when air drying at room T was used and 60.5 mg/g d.w. in the case of hot air drying. It was found out that extracts of freeze-dried leaves have a significantly lower content of oleuropein compared to other methanol extracts. Methanol extracts of *Istrska belica* are richer in oleuropein, whereas *Leccino* methanol extracts have higher content of other phenols, such as hydroxytyrosol glucoside (both isomers 1 and 2) and oleuroside. In lyophilized samples, some isomers appear that are not detectable in air-dried leaves, such as the presence of secologanoside (Sec), which was detected only in methanol extracts of freeze-dried olive leaves. Şahin et al. [1] performed SFE extraction modified with ethanol at 300 bar and 50 °C. The extraction yield of oleuropein in this case was 2.9 mg/g d.w. but was much higher around 100 mg/g d.w. at a temperature of 100 °C. In our study, mild conditions of pressure and temperature were used, but still, 13.6 mg/g oleuropein was determined. Baldino et al. performed supercritical antisolvent extraction (SAE) with ethanol, operating at different pressures between 100 and 200 bar and temperatures between 35 °C and 60 °C, with the highest oleuropein content in ethanol solution being about 20% *w*/*w* [37].

It was found that eutectic solvents (DESs) are also beneficial for the extraction of some polyphenols from olive leaves. The highest concentration of 45 mg/g d.w. was obtained in *Istrska belica* extracts in the case of air drying at room T, whereas drying at 105 °C resulted in a concentration of 43.1 mg/g d.w. Oleuropein content in *Leccino* extract was comparable in the case of hot drying at T = 105 °C, whereas in the case of air drying, the concentration was slightly lower (about 30.2 mg/g d.w.). A special feature was detected is the presence of oleuropein glucoside with *m/z* 701 in eutectic extract. Something similar was reported by Alañón et al. [22]. During hot air drying, the high temperatures place great stress on the cell walls, and drying makes the release of phenolic compounds into the solvent easier. This could be reason for higher concentrations of oleuropein during *Leccino* extraction with DES. A similar effect was explained by Ahmad-Quasem et al. [38]. They also reported that drying and freezing have a significant effect on the content of phenols in olive leaf extract.

In our study, methanol extraction proved to be most effective for extraction of oleuropein. The highest determined concentration was 77.7 mg/g d.w., followed by ultrasonic extraction with deep eutectic solvent, with the highest concentration of 45 mg/g d.w. The lowest oleuropein contents were confirmed in the samples obtained by modified supercritical extraction of freeze-dried leaves: only 13.6 mg/g d.w.

Previous studies have proven that freeze drying is not always an appropriate method for improving the extraction of phenolic compounds from olive leaves. On the other hand, it is known that ice crystals formed within the plant matrix during freezing can improve extraction efficiency in other materials. Ice crystals can damage the cell structure and allow for extraction of inner components; solvent contact is easier, and consequently, extraction is more effective [20]. Interestingly, SCE extracts of freeze-dried leaves have higher oleuropein contents than extracts of dried leaves, which, in other cases, showed the highest levels of active substances. For instance, supercritical extracts from-freeze dried *Istrska belica* leaves had an oleuropein concentration of 13.3 g/g d.w., which is higher than that in methanol and DES extracts from the same material, where concentrations of 4.9 mg/g d.w and 0.9 mg/g d.w. were determined. As previously reported by Afaneh et al., degenerative enzymes, such as β-glucosidase, could be released during lyophilization, which could significantly reduce oleuropein content [31]. Lyophilization results in the formation of certain isomers that are not observed in other extracts to such an extent, such as luteolin-7-glucoside, with a highest concentration of between 1.3 and 2 mg/g d.w determined in extracts from freeze-dried leaves using DES and scCO_2_ extraction. Nevertheless, we used concentrations of oleuropein in extracts from freeze-dried leaves of both cultivars, which are significantly lower than if drying at room T or at T = 105 °C. Based on our research, freeze drying does not seem to be an adequate drying method for improving the extraction of oleuropein from *Istrska belica* or *Leccino* olive leaves. It can be seen that the polarity of the solvent also affects the yield of polyphenols. The highest yield of polar components, i.e., polyphenols, was observed when methanol was used as a solvent. The yield of polyphenols was lower when using scCO_2_ with ethanol due to the lower polarity of the solvent. In addition to the drying processes, the extraction techniques used, and the solvents chosen, the antioxidant activity and the olive polyphenol content are certainly influenced by the type of cultivar itself, the age of the tree, and the location.

## 4. Materials and Methods

### 4.1. Materials

Olive leaves of *Istrska belica* and *Leccino* cultivars were obtained in late February in Izola (Slovenia) region. The leaves of the outer biennial branches of regularly pruned trees were sampled. Cultivar identification was first performed based on the description of the specific plant; however, in addition, molecular markers were used. For the purpose of our research, the cultivars from Izola were evaluated, whereas the oligonucleotide profile was not determined. Paternity analysis based on microsatellite markers was used for genotyping and identification [39]. Basic data about harvesting, sampling, and drying olive leaves are presented in Table 5.

Carbon dioxide (CO_2_) (PubChem CID: 280), purity 99.5% vol., was delivered by Messer (Ruše, Slovenia); ethanol abs. (PubChem CID: 702), purity 99.9% vol., was supplied by Carlo Erba (Milan, Italy); and methanol (PubChem CID: 329755069), purity 99.9% vol., glycerol (PubChem CID: 24895216), purity ≥ 99.0% vol., glycine, purity 99.9% vol., were supplied by Merk (Darmstadt, Germany). Standards of olive polyphenols were supplied by Sigma Aldrich (St. Louis, MI, USA).

### 4.2. Methods

#### 4.2.1. Drying

Three leaf-drying techniques were performed on olive leaves. Air drying at room temperature was performed at a constant temperature of 20 °C for 10 days in a dark place. Another procedure was drying at a temperature of 105 °C for 90 min. Another type of drying was lyophilization or freeze drying preformed in a Kambič LIO 2000 PNS freeze dryer under vacuum conditions. The dried leaves were placed in paper bags and ground before extraction if necessary. Approximately 2 g of dried sample was weighed, and the moisture content was determined by a Halogen HX204/M moisture analyzer.

#### 4.2.2. Ethanol-Modified Supercritical Extraction

SC extraction (SCE) experiments were performed in a semicontinuous high-pressure flow unit previously described in the literature [40]. The high-pressure vessel was loaded with approximately 10 to 15 g of ground material (particle size, 2.0 mm to 3.0 mm) and placed in a water bath heated to the desired temperature (T = 40 °C). Extraction was performed with supercritical CO_2_ and ethanol (EtOH) as cosolvents. EtOH was pumped continuously using a high-pressure pump with a flow rate of 2 mL/min. Pressurized CO_2_ was introduced in the autoclave from the gas cylinder. Pressure was kept constant at 140 bar during the entire experiment and was regulated by a control valve. The extract and the solvent were collected in glass tubes. The longest time of extraction was 120 min (time of one fraction cycle, 30 min). Each solution from individual fractions was transferred to an evaporation flask, and the solvent was evaporated. The extract was stored in a freezer at −10 °C until analysis.

#### 4.2.3. Methanol Extraction

Approximately 1 g of ground leaf (particle size, 2.0 mm to 3.0 mm) sample was weighed and introduced into the flask, and 50 mL of preheated methanol (60 °C) was added according to the method of European Pharmacopeia 10.0 [27]. The extraction was carried out for one hour at a temperature of 60 °C and constant stirring. The material was separated from the solution, and the solvent was evaporated afterward. 

#### 4.2.4. Ultrasound Extraction with Deep Eutectic Solvent (DES)

The preparation of deep eutectic solvent (DES) was summarized by V. Athanasiadis et al. [20]. Glycerol, glycine, and water were used for preparation of DES in molar ratios of Gli:Gly:H_2_O = 7:1:9 at temperatures of 80 °C to 90 °C. Approximately 1.5 g of ground olive leaves (particle size, 2.0 mm to 3.0 mm) was weighed and introduced into an Erlenmeyer flask. Then, 10 mL of DES was added. The reaction mixture was then immersed into an ultrasonic bath (40 kHz). Extraction was performed at a constant temperature of 50 °C for 1 h. Then, the mixture was centrifugated at 6000 rpm and filtered afterward. Filter residue was discarded, and the extract was frozen until further HPLC analysis.

#### 4.2.5. DPPH Radical Scavenging Assay

Radical scavenging activity of extracts was measured using the stable radical DPPH (2,2-diphenyl-picryl-hydrazil) reagent. Extract solutions of 1 mg/mL in methanol were prepared. Then, 3 mL of 0.06 mM DPPH solution was pipetted into dark bottles, and 0.77 µL of extract solution was added. The sample was stirred and thermostated at room temperature for 15 min. Afterward, the absorbance at 515 nm was measured using a UV-VIS spectrophotometer (CARY 50 UV-VIS). A reference solution was prepared by adding 0,77 methanol to the DPPH solution and measuring the absorbance immediately. Antioxidant activity was calculated accordingly and expressed as % inhibition [41].

#### 4.2.6. HPLC—DAD-MS/MS Analysis 

For identification and determination of active compounds (polyphenols; Table 6), the HPLC (Infinity 1260 Agilent Technologies, Waldbronn, Germany) interfaced with a triple quadrupole mass spectrometer (QqQ MS/MS 6420 system; Agilent Technologies, Singapore). The HPLC was equipped with a standard electrospray ionization source (model G1948B), a degasser (model G4225A), a binary gradient pump (model G1312A), a thermoautosampler (model G1329B), a column oven (model G1316A), and a diode-array detection system (model G4212B).

Chromatographic separation of the compounds was performed on an analytical column Luna, PFP (2), 100 Å, 250 mm × 4.6 mm, 5 µm particle size.

The elution gradient consisted of mobile phase A (water with addition of 0.1 vol.% of formic acid) and mobile phase B (acetonitrile:methanol (50:50 *V*/*V*) with addition of 0.1 vol.% of formic acid). The flow rate was 1 mL/min using gradient a program as follows: 0 min 4% B, 40 min 50% B, 45 min, 60% B, 60 min 100% B, 70 min 100% B, and at 72 min, back to 4% B. Samples subjected to the analyses were prepared by weighing approximately 20 mg of the extract in a 10 mL volumetric flask and diluting it with MeOH:H_2_O (*v*/*v*) up to 10 mL. Prepared samples were filtered through a 0.2 µm syringe filter and injected (volume of 20 µL) into the system. All measurements were performed in duplicate (n = 4), and results are expressed as mean value ± SD of mg∙g^−1^. All the identified compounds were quantified on the basis of oleuropein response at 280 nm. The calibration curve for oleuropein (Sigma-Aldrich, Buchs, Switzerland, Cat. no. 92167) was linear, in a range from 0.03 to 5.0 µg per injection volume, R^2^ = 0.99991. The limit of detection was established at 0.01 mg∙g^−1^ (S/N = 3). All the identified compounds were confirmed according to their molecular peak or/and transition according to the confirmation data from Table 6.

The MS and MS/MS data were acquired in negative ionization mode using the following optimized conditions: sheath gas temperature (300 °C), flow (11 L/min), nebulizer pressure (241.32 kPa), capillary voltage (4.0 kV), cell acceleration voltage (7 V), fragmentor voltage (80–220 V), collision energy (15–35 V (MS/MS)) [42]. 

## 5. Conclusions

In our study, leaves of two different olive cultivars, *Istrska belica* and *Leccino*, were investigated. Our research was focused on three different types of extraction procedures. Drying of olive leaves before extraction is required to achieve high oleuropein content. It also prevents microbial fermentation and further degradation. Based on DPPH analysis of methanol and supercritical extracts, we can conclude that olive leaf extracts, especially from *Istrska belica*, have extremely high antioxidant efficiency due to the synergistic action of olive biophenols, such as oleuropein, verbascoside, luteolin-7-O-glucoside, and others. High DPPH activity is not necessarily associated with high concentrations of certain active components. We successfully determined several polyphenols, such as hydroxytyrosol glucoside, methoxy-oleuropein, apigenin, apigenin-7-glucoside, glucoside-oleuropein, hydroxy oleuropein, ligstroside, luteolin, luteolin-7-glucoside, oleuropein, oleuroside, secologanoside, hydroxytyrosol, and verbascoside. The highest concentration of oleuropein was reached when air drying at room T was used, followed by air drying at T = 105 °C and freeze drying. Combining the drying process and the type of extraction, it can be summarized that drying *Istrska belica* and *Leccino* at room T by use of conventional methanol extraction indicates the highest concentration of oleuropein of 77.7 mg/g d.w. and 66.1 mg/g d.w., respectively. In general, *Istrska belica* is richer in polyphenols compared to *Leccino*. This can be seen from the DPPH analysis and from the qualitative and quantitative phenolic profile. Freeze drying causes chemical conversions in the leaves, resulting in conversions of the active compounds, which causes concentrations that are higher, lower, or not detected. Freeze drying does not seem to be an appropriate method for pretreatment of material before extraction of oleuropein from *Istrska belica* and *Leccino* olive leaves. In general, the most efficient technique for extraction of oleuropein for both cultivars was conventional extraction with methanol, followed by extraction with DES and modified supercritical extraction. The most appropriate method is drying at room T.

## Figures and Tables

**Figure 1 plants-11-00865-f001:**
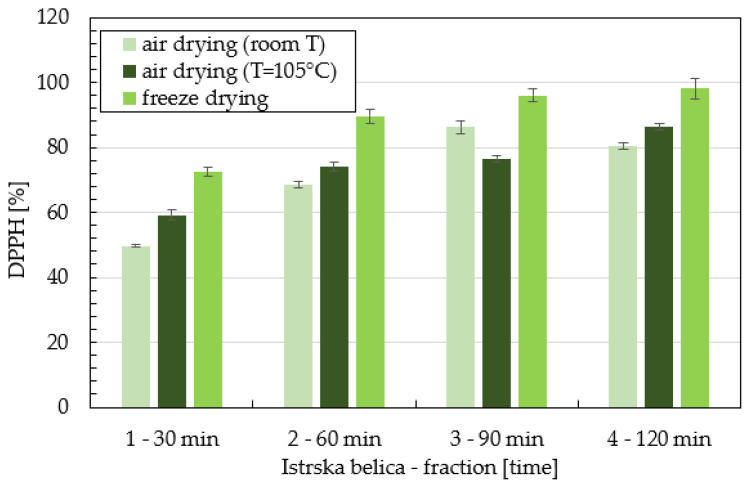
Comparison DPPH activity of extracts of *Istrska belica* in % obtained by SCE extraction with ethanol at different fractions and with different methods of drying. Data are means ± SD from three replicates.

**Figure 2 plants-11-00865-f002:**
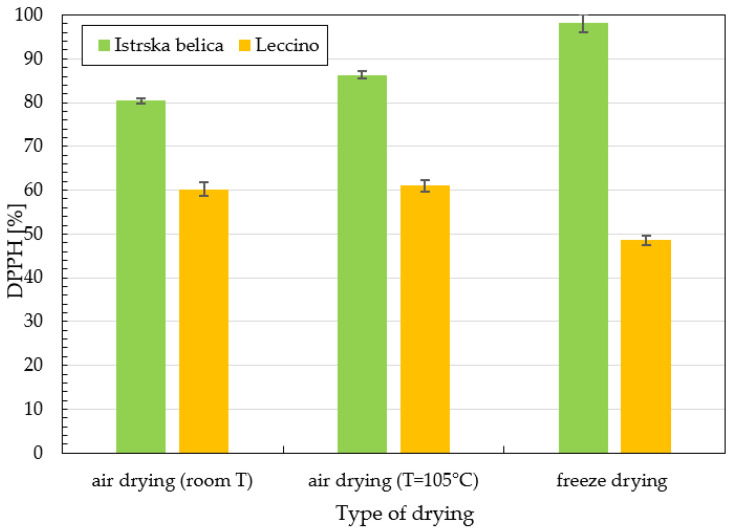
Comparison of DPPH activity in % using SCE extraction after 120 min with ethanol for *Leccino* and *Istrska belica* with different methods of drying. Data are means ± SD from three replicates.

**Figure 3 plants-11-00865-f003:**
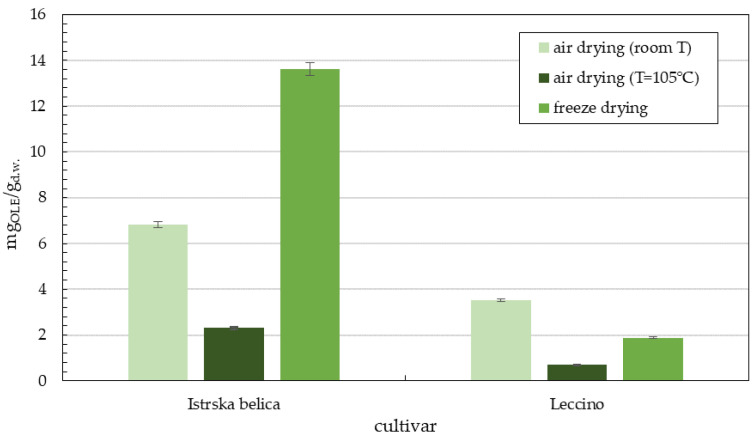
Comparison of oleuropein content (mg_OLE_/g d.w.) in extracts of *Istrska belica* and *Leccino* obtained by SCE extraction after 90 min for different methods of drying. Data are means ± SD from three replicates.

**Figure 4 plants-11-00865-f004:**
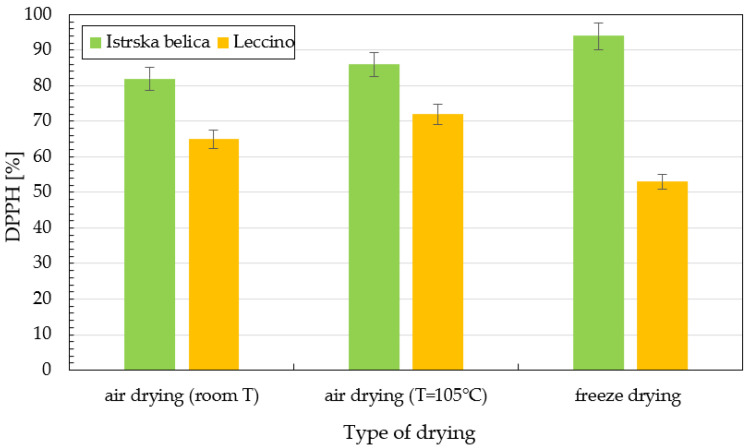
Comparison of DPPH activity in % using extraction with methanol for *Leccino* and *Istrska belica* with different methods of drying. Data are means ± SD from three replicates.

**Figure 5 plants-11-00865-f005:**
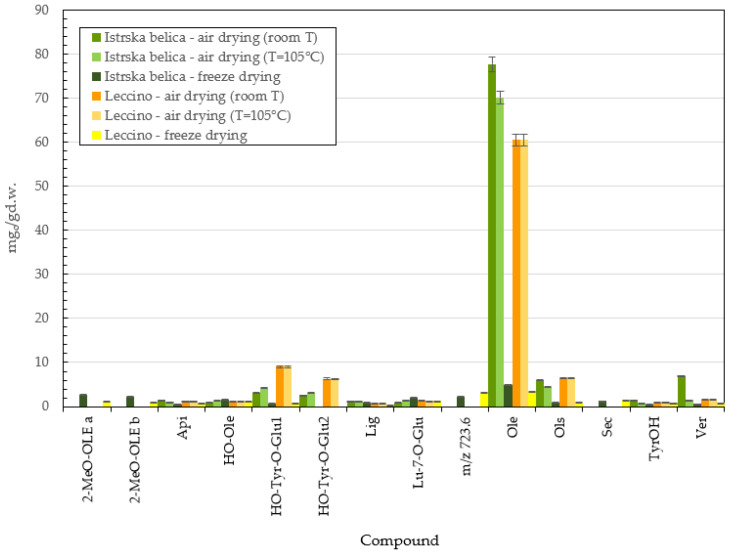
Content in mg_c_/g d.w. of active compounds of *Istrska belica* and *Leccino* methanol extracts with different methods of drying. Data are means ± SD from three replicates.

**Figure 6 plants-11-00865-f006:**
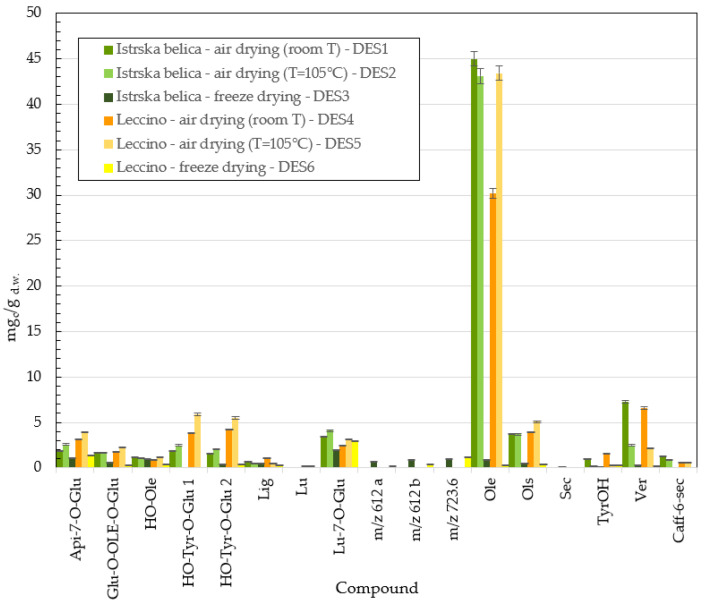
Content of active compounds (mg_c_/g d.w.) in eutectic extracts of *Istrska belica* and *Leccino.* Data are means ± SD from three replicates.

**Table 1 plants-11-00865-t001:** Moisture content in leaves using different types of drying.

Cultivar	Type of Drying	Time of Drying	Water Content [%]
*Istrska belica*	air–room temperature	10 days	4.98 ± 0.25
	air–dryer T = 105 °C	90 min	3.42 ± 0.17
	freeze–dryer	until the constant weight	6.24 ± 0.31
*Leccino*	air–room temperature	10 days	4.87 ± 0.22
	air–dryer T = 105 °C	90 min	3.39 ± 0.21
	freeze dryer	until the constant weight	6.37 ± 0.33

±SD of three independent experiments.

**Table 2 plants-11-00865-t002:** Extraction yield of *Istrska belica* and *Leccino* cultivars.

Sample ID	Type	Extraction Time [Min]	Fraction Yield [%]	Extraction Yield [%]
SC *_1	*Istrska belica*, air–room T	30	4.597	
SC_2		60	3.519	12.6
SC_3		90	3.380	
SC_4		120	1.065	
SC_5	*Leccino* air–room T	120	/	6.8
SC_6	*Istrska belica* air–dryer T = 105 °C	30	2.964	
SC_7		60	1.393	7.5
SC_8		90	1.572	
SC_9		120	1.004	
SC_10	*Leccino* air–dryer T = 105 °C	120	/	1.2
SC_11	*Istrska belica* freeze–dryer	30	/	
SC_12		60	6.047	9.7
SC_13		90	3.087	
SC_14		120	0.457	
SC_15	*Leccino* freeze–dryer	120	/	4.3

* supercritical.

**Table 3 plants-11-00865-t003:** Concentration of active compound (C) in mg_c_/g d.w. for individual fractions of ethanol-modified supercritical extraction of *Istrska belica* with different types of drying.

Type	*Istrska belica*										
	Air Drying, Room T				Air Drying, T = 105 °C				Freeze Drying		
Sample ID	SC_1	SC_2	SC_3	SC_4	SC_6	SC_7	SC_8	SC_9	SC_12	SC_13	SC_14
Time SCE (min)	30	60	90	120	30	60	90	120	60	90	120
Compound	mg_c_/g d.w.										
HO-Tyr-O-Glu ^1^	n.d. ^16^	0.015 ± 0.001	0.092 ± 0.003	0.027 ± 0.001	0.036 ± 0.001	0.038 ± 0.0008	0.055 ± 0.0012	0.056 ± 0.001	n.d.	0.338 ± 0.007	0.237 ± 0.005
2-MeO-OLE a ^2^	0.396 ± 0.01	0.201 ± 0.006	0.108 ± 0.004	0.065 ± 0.002	0.096 ± 0.002	0.0271 ± 0.0006	0.0272 ± 0.0007	0.0181 ± 0.0004	0.0021 ± 0.0001	n.d.	0.0171 ± 0.0004
2-MeO-OLE b ^3^	0.259 ± 0.007	0.143 ± 0.004	0.054 ± 0.002	0.028 ± 0.001	n.d.	n.d.	n.d.	n.d.	n.d.	0.162 ± 0.003	0.037 ± 0.001
Api ^4^	0.015 ± 0.001	0.014 ± 0.004	0.016 ± 0.001	0.01 ± 0.0004	0.0192 ± 0.0004	0.00621 ± 0.00013	0.0051 ± 0.0001	0.0040 ± 0.0001	0.00201 ± 0.00004	0.035 ± 0.001	0.0152 ± 0.0003
Api-7-O-Glu ^5^	0.335 ± 0.01	0.231 ± 0.007	0.208 ± 0.007	0.13 ± 0.005	0.0451 ± 0.001	0.0473 ± 0.0009	0.0581 ± 0.0012	0.054 ± 0.001	n.d	0.492 ± 0.010	0.300 ± 0.006
Glu-O-OLE-O-Glu ^6^	0.176 ± 0.005	0.094 ± 0.003	0.060 ± 0.002	0.036 ± 0.001	0.0261 ± 0.0006	0.0181 ± 0.0004	0.0191 ± 0.0004	0.0163 ± 0.0003	n.d.	0.119 ± 0.002	0.058 ± 0.001
HO-Ole ^7^	n.d.	0.014 ± 0.001	0.011 ± 0.001	0.032 ± 0.001	n.d	0.0141 ± 0.0003	0.0171 ± 0.0004	0.0161 ± 0.0003	0.012 ± 0.0003	0.148 ± 0.003	0.109 ± 0.002
Lig ^8^	0.171 ± 0.005	0.142 ± 0.005	0.15 ± 0.005	0.091 ± 0.003	0.0421 ± 0.001	0.042 ± 0.001	0.044 ± 0.001	0.0331 ± 0.0007	0.0011 ± 0.0002	0.226 ± 0.005	0.129 ± 0.003
Lu ^9^	0.021 ± 0.001	0.016 ± 0.003	0.012 ± 0.001	0.018 ± 0.001	0.0251 ± 0.001	0.0084 ± 0.0002	0.0081 ± 0.0002	0.0062 ± 0.0001	0.0010 ± 0.0002	0.041 ± 0.001	0.0190 ± 0.0004
Lu-7-O-Glu ^10^	0.067 ± 0.002	0.087 ± 0.002	0.199 ± 0.239	0.127 ± 0.005	0.0261 ± 0.048	0.043 ± 0.001	0.064 ± 0.002	0.079 ± 0.002	n.d.	1.092 ± 0.023	0.712 ± 0.015
Ole ^11^	2.229 ± 0.06	3.513 ± 0.11	6.834 ± 0.022	4.288 ± 0.15	1.9281 ± 0.005	1.969 ± 0.041	2.303 ± 0.048	2 ± 0.042	0.0041 ± 0.0001	13.603 ± 0.286	8.357 ± 0.18
Ols ^12^	0.324 ± 0.009	0.393 ± 0.012	0.627 ± 0.001	0.392 ± 0.014	0.20211 ± 0.0003	0.164 ± 0.003	0.181 ± 0.004	0.145 ± 0.003	0.0031 ± 0.0001	1.059 ± 0.022	0.625 ± 0.013
Sec ^13^	n.d.	0	0.037 ± 0.017	0.017 ± 0.001	0.0121 ± 0.002	0.0092 ± 0.0002	0.0141 ± 0.0003	0.0150 ± 0.0003	n.d.	0.090 ± 0.002	0.065 ± 0.001
TyrOH ^14^	0.23 ± 0.006	0.372 ± 0.011	0.488 ± 0.005	0.305 ± 0.011	0.0871 ± 0.002	0.0272 ± 0.0006	0.0202 ± 0.0004	0.012 ± 0.0002	n.d.	0.078 ± 0.002	0.034 ± 0.001
Ver ^15^	0.029 ± 0.001	0.172 ± 0.005	0.146 ± 0.239	0.08 ± 0.003	0.0621 ± 0.05	0.0234 ± 0.0005	0.0122 ± 0.0003	0.0091 ± 0.0002	n.d.	0.176 ± 0.004	0.107 ± 0.002

^1^ Hydroxytyrosol glucoside, ^2^ methoxy-oleuropein a, ^3^ methoxy-oleuropein b, ^4^ apigenin, ^5^ apigenin-7-glucoside, ^6^ glucoside-oleuropein, ^7^ hydroxy oleuropein, ^8^ ligstroside, ^9^ luteolin, ^10^ luteolin-7-Glucoside, ^11^ oleuropein, ^12^ oleuroside, ^13^ secologanoside, ^14^ hydroxytyrosol, ^15^ verbascoside; ^16^ not defined; SD ± means the standard deviation.

**Table 4 plants-11-00865-t004:** Concentration in mg_c_/g d.w. for individual fractions of ethanol-modified supercritical extraction of *Leccino* as a function of different drying procedures.

Type	*Leccino*		
	Air Drying, Room T	Air Drying, T = 105 °C	Freeze Drying
Sample ID	SC_5	SC_10	SC_15
Time SCE (min)	120	120	120
Compound	mg_c_/g d.w.		
HO-Tyr-O-Glu ^1^	0.119 ± 0.002	0.032 ± 0.001	0.079 ± 0.002
2-MeO-OLE a ^2^	n.d. ^16^	n.d.	0.065
2-MeO-OLE b ^3^	n.d.	0.0241 ± 0.0005	n.d.
Ap ^4^	0.046 ± 0.001	0.0050 ± 0.0001	1.330 ± 0.028
Api-7-O-Glu ^5^	0.100 ± 0.002	0.0260 ± 0.0005	0.860 ±0.0005
Glu-O-OLE-O-Glu ^6^	0.070 ± 0.001	0.013 ± 0.0003	0.271 ± 0.0057
HO-Ole ^7^	n.d.	n.d.	0.266 ± 0.0060
Lig ^8^	0.080 ± 0.002	0.018 ± 0.0004	0.078 ± 0.0004
Lu ^9^	0.026 ± 0.001	0.0101 ±0.0002	0.439 ± 0.0002
Lu-7-O-Glu ^10^	0.058 ± 0.012	0.0111 ± 0.0002	1.418 ± 0.0002
Ole ^11^	3.522 ± 0.074	0.707 ± 0.015	1.886 ± 0.039
Ols ^12^	0.517 ± 0.011	0.093 ± 0.002	0.494 ± 0.01
Sec ^13^	0.066 ± 0.001	0.009 ± 0.0002	n.d.
TyrOH ^14^	0.814 ± 0.017	0.025 ± 0.0005	0.098 ± 0.002
Ver ^15^	0.155 ±0.003	0.021 ± 0.0004	0.117 ± 0.002

^1^ Hydroxytyrosol glucoside, ^2^ methoxy-oleuropein a, ^3^ methoxy-oleuropein b, ^4^ apigenin, ^5^ apigenin-7-glucoside, ^6^ glucoside-oleuropein, ^7^ hydroxy oleuropein, ^8^ ligstroside, ^9^ luteolin, ^10^ luteolin-7-glucoside, ^11^ oleuropein, ^12^ oleuroside, ^13^ secologanoside, ^14^ hydroxytyrosol, ^15^ verbascoside; ^16^ not defined SD ± means the standard deviation.

**Table 5 plants-11-00865-t005:** Data on harvested olive leaves.

Olive Cultivar	Location	Harvesting Month	Sampling Air Temperature	Age of Trees	Drying Method
*Istrska belica*	45°31′22.8″ N Izola, Slovenia	February	11 °C	6 years	air at room temperature dryer at T = 105 °C lyophilization
*Leccino*	13°39′47.2″ E Izola, Slovenia				

**Table 6 plants-11-00865-t006:** Active compounds.

Active Compound	Confirmation
Oleuropein (Ole)	[M-H^−^] 539, MS/MS (539/275)
Verbascoside (Ver)	[M-H^−^] 623, MS/MS (623/161)
Oleuroside (Ols)	[M-H^−^] 539, MS/MS (539/275)
Ligstroside (Lig)	[M-H^−^] 523
Hydroxy oleuropein (HO-Ole)	[M-H^−^] 555
Hydroxytyrosol glucoside [HO-Tyr-O-Glu]	[M-H^−^] 315
Hydroxytyrosol (TyrOH)	[M-H^−^] 153, MS/MS (153/123)
Luteolin-7-Glucoside	[M-H^−^] 447, MS/MS (447/285)
Apigenin-7-Glucoside (Api-7-O-Glu)	[M-H^−^] 431, MS/MS (431/268)
Secologanoside (Sec)	[M-H^−^] 389
Luteolin (Lu)	[M-H^−^] 285, MS/MS (285/133)
Apigenin (Api)	[M-H^−^] 269 (269/151)

## Data Availability

Not applicable.

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
