# Peer review of "The Effect of Drying Methods and Extraction Techniques on Oleuropein Content in Olive Leaves"

_plants, 2022, doi:10.3390/plants11070865_

Round 1

Reviewer 1 Report

Please find my comments below:

  • Paragraph lines 52-57: I would expand this paragraph to discuss NDES solvents.
  • Lines 88 and 390: There should be a period in the numbers, not a comma.
  • Line 99 and Table 2: What was the extraction time for the Leccino variety? Was it 90 or 120 minutes?
  • Tables 3 and 4: Please add an explanation of compound abbreviations below the tables.
  • Line 160: Please correct the font size.
  • Why did the authors choose only one eutectic solvent? Why this one? Maybe another one would get better results?
  • Line 220:” Leccino “should be written in italic font.

Reviewer 2 Report

Dear Authors,

In the article „The Effect of Extraction Techniques and Drying Methods on
Oleuropein Content in Olive Leaves” you are presenting the very interesting conception of olives oil waste management. The processing of olive leaves due to obtain extracts rich in biologically active compounds seem to be much more sensible than composting. Although, after reading the manuscript I have some suggestions and comments.

Title
“Drying methods” should be mentioned before “extraction techniques”, since the material was dried and then the extraction was carried on.

Abstract
In lines 16-17 and 22-23 you repeated the same information.

Key words
“drying methods” seem to be not precise. Please, specify methods or leave only “drying”

Introduction
The most important deficiency of ”Introduction” is the lack of description
1. problems concerning the influence of drying conditions on the chemical composition of plant material, justifying the examination of different drying methods you have made
2. differences in extraction methods, explaining why have you chosen yours.

line 27
…in addition to oil…
better would sound: during the olive oil production
line 28
…present…?
Did you mean “represent”?
line 30
… deposited in nature…
What do you mean? Composting?
line 34
…around 2-3%...
it should be in brackets, or separated from the text by comas
line 37
without a completed chemical characteristics
lines 39 and many more cases
leaf extract or leaves extract
use one form
additives (should be plural form)
lines 47,48 and 49
You repeated “It was found out…”

Results
2.1. Water content or humidity of …
Table 1
Standard deviation is missing. In the title should be information that you present water content in material dried by different drying methods.
line 106
If you compare two things you should write: “higher” not “highest”.
It is not necessary to explain the unit of components concentration as many times as you do it. It should be described in the section “Methods” only, not in every table and figure title.
It is not well explained why did you carried out the extraction of “Istska belica” using different time of process (30, 60 and 90 minutes) while the extraction of “Leccino” – not.
line 219
…the opposite.. what? Phenomena?
line 235
“drying methods” not “drying types”. Celsius should have the capital letter.
lines 233-235
First you have dried leaves and then you performed extraction. Keep this order, please.
line 236
This information should begin the “Discussion” section. Then you should explain the reason of carrying out drying by this tree methods you have chosen.

In the section “Discussion” you provide information that should be placed in the section “Results”. In “Discussion” you should try to explain the reason of the results you observed and compare them to others authors. Separate this two sections according to logic.

Conclusions
The last sentence should be paste just after information about drying. In the present form you have info about drying then extraction and then again drying. It is not logically.

Best regards!

Reviewer 3 Report

This work by D.C. Andrejc et al. Is an interesting summary of the research on the influence of the extraction method on the oleuropein content in olive leaves.

The article is well written, the research is well planned. However, in my opinion, it needs some corrections.

(66) After the leaves samples ... - this sentence should be in the M&M section

Results
Table 1, last column - mean. Average of what? How many samples?
Table 2. The table caption and the heading of the last column should be corrected (the penultimate and the last column have the same headings).
Error bars are shown in the bar charts. What statistical analyzes were used?
DPPH results are shown on a % scale. I recommend converting to mg of scavenged radical. This enables comparisons with the results of other studies.
HPLC results.
Has the method been validated? Please enter validation parameters.
How many repetitions were performed? Has a statistical analysis been used? (Tukey's test? please indicate in the table 3 groups that differ statistically with the letters a, b, c, ...)
Table 3 should contain the same number of significant figures for all results. Measurements should be reported with the same accuracy.

Reference [16] - why is such an old version of the European Pharmacopoeia used?

M&M
How were specific olive varieties identified? Voucher number?

Please state the methods of statistical evaluation of the results used.

Reviewer 4 Report

I commend the authors of the manuscript titled “The Effect of Extraction Techniques and Drying Methods on  Oleuropein Content in Olive Leaves” for their work on  the influence of different drying methods and extraction procedures on the active substance content in olive leaves.

Before this manuscript is published, there are several things need to be addressed or corrected:

  • In the abstract: line 16, “this “ should be added before “manuscript”, the word “manuscript” should be replaced by “research”.

Line, 20, “the antioxidant activity was determined” not “ antioxidant activity to”

The English should be revised in this manuscript by native English speaker.

You need to mention the major polyphenols determined in the leaves.

The result or your recommendation for the best method should be addressed in the abstract and conclusion.

  • In the introduction:
  • The introduction is comprehensive and no changes need to be done there

  • In the results
  • Table 1 and 2 need to add standard deviation or standard error values.
  • Figure 1 significant difference should be shown.
  • Table 3 need to be converted to landscape page style to be more readable .
  • In chapter 2.2. the title should be “results of “ not “on” also in chapter 1.1.
  • Chapter 2.4. the title should be “results of “
  • Figure 6 is not readable and need to be enlarged or divided
  •  
  • In the discussion part: I feel the discussion is long and need to be shortened
  • In the material and methods:

 section 4.1 :  you need to determine the voucher of the plant or species used and/or the person who identified the species or cultivar.

  • The conclusion, the conclusion need to be shortened to single paragraph.

Reviewer 5 Report

Andrejč et al. describe in their manuscript different drying methods for olive leaves of two different varieties and different extraction methods and their influence on the yield of oleuropein, a major compound of olive leaves. Based on their experiments the authors conclude that drying at room temperature and extraction with methanol are most appropriate for the isolation of oleuropein. Altogether 25 references are used.

There are several issues which should be addressed by the authors:

General Aspects

In the introduction some remarks on the health benefits of olive leaves and oleuropein should be added. The introduction does not give a good explanation why oleuropein is a valuable compound and why this compound and not an other compound is isolated from olive leaves.

The chapter „discussion“ summarizes the findings of this study, but in this part there should also be a discussion of these findings in relation to other publications. There are numerous publications on the isolation and purification of oleuropein available which should be considered in the discussion.

Why did you use different extractions methods for Istrska belica and Leccino, i.e. 4 x 30 min extraction vs. 1 x 90 min (see table 2) for the extraction with supercritical CO2? The results of the yields (see Table 2) and also the results of the DPPH-testing (see Figure 2) and the content of oleuropein (see Figure 3) can therefore not be compared properly.

A very basic aspect is the grade of comminution of the herbal drug. For the extraction with supercritical carbondioxide and the ultrasound extraction the (whole) dried leaves are used. However, based on the monograph of the European Pharmacopoeia, the comminuted leaves („ground leaf samples“, see 4.2.3) are used. This makes an enormous difference in the yield, as comminution breaks the cell walls and facilitates the extraction process. Thus it is not a surprise that the yield of oleuropein is the highest when the comminuted herbal material is extracted with methanol.

Figure 5 and 6: The values for each compound should be added together with an example of the chromatogram as supplementary material.

A list of abbreviations should be added (e.g. „T“, „SCE“ etc.)

Further Aspects

Line 86: according to the European Pharmacopoeia, monograph 1878 on olive leaves, the moisture content of less than 10.0 % is related to the exact conditions given in this monograph, but not to any other conditions as used here. This should be mentioned in the text.

Line 90 / Table 1: the time of drying of each method should be added!

Line 99: for the Leccino variety this should be „90“ min (not 120 min, see table 2)

Table 3: regarding freeze drying, SC_11 related to 30 min, SC_12 to 60 min, and SC_13 to 90 min. However, results for SC_14 (120 min) are missing. Please add these data!

Table 3: Some values are zero („0“). Please present the detection limit and/or the limit of quantification of you HPLC method!

Line 169 and reference 16: The current version of the European Pharmacopoeia is 10.0, but not 5.0 from 2005 (although the methods are most likely still the same).

Line 310-311 and Figure 5 and 6: Two isomers of Luteolin-7-O-glucoside? This compound is definitely only one compound. According to Table 6 the fragmentation patterns of the „two isomers“ are different, indicating two different compounds. Please check again!

Line 384 / 4.2.6 The instrumentation of the HPLC-DAD is described, but not of the LC MS/MS. Please add these details!

Table 6: There are some more compounds mentioned in Figure 5 and 6, which are not part of Table 6. Please add these compounds and their mass fragmentation which was used for their identification!

To sum up, the isolation of oleuropein and its optimization is an interesting topic for a publication. However, serveral aspects should thoroughly be reconsidered by the authors and several changes need to be performed.

Round 2

Reviewer 3 Report

The Authors responded to all questions and suggestions. The manuscript has been corrected and may be accepted for publication.

Author Response

No comments from the reviewer.

Reviewer 4 Report

Accepted for me. 

Author Response

No comments from the reviewer.

Reviewer 5 Report

Andrejč et al. have prepared a revised form of their manuscript on different drying and extraction methods and the corresponding oleuropein contents based on the comments of the reviewers. Regarding my review not all of my remarks were considered. Thus there are still several issues which need to be addressed by the authors before acceptance of the manuscript:

The introduction regarding the health benefits of olive leaves and oleuropein, respectively, is still extremely general. Further publications on this aspect should be added in the introduction in order to show the audence that oleuropein is really an important and valuable compound which should be isolated from olive leaves.

The chapter „discussion“ still summarizes the findings of this study. The authors have now added two publications, i.e. one in relation to the antioxidant activities of olive leave extracts and one on the extraction yield. However, in the part „discussion“ a broad discussion on the yield of oleuropein isolated from other olive tree varieties or if possible of the varieties Leccino and Istrska belica should be added. As mentioned in my previous review there are numerous publications on the isolation and purification of oleuropein available which should be considered in the discussion.

A very basic aspect is the grade of comminution of the herbal drug. Based on the chapter „material and methods“ comminuted material was used for SCE and methanol extraction. At least there is no hint in chapter 4.2.4 that comminuted plant material was used for eutectic solvent extraction. Please check again!

Table 3: regarding freeze drying, SC_11 is related to 60 min, SC_12 to 90 min, and SC_13 to 120 min. However, these codes are not identical with the codes given in table 2 (in table 2: SC_12 is related to 60 min, SC_13 to 90 min, SC_14 to 120 min). Therefore it is difficult to follow. Please correct this!

Table 4: for SCE of the Leccino variety, SC_14 is attributed to freeze drying and 120 min SCE. However, the sample SC_14 in table 2 is attributed to variety Istrska belica, extraction time 120 min. Please correct the codes! This is really confusing!

Table 3: The authors have determined the contents of the natural compounds on the basis of oleuropein response at 280 nm. It is impossible to use the same response factor for such different compounds like flavonoids (e.g. apigenin), polyphenolic seco-iridoids (i.e. oleuropein derivatives) and hydroxytyrosol. This is analytically much too simple although these compounds have an aromatic system in common! The determination of the compounds should be modified!

As mentioned in my previous review, the isolation of oleuropein and its optimization is an interesting topic for a publication. However, serveral aspects still need a reconsideration by the authors and several changes need to be performed.
